# ESA, Iron Therapy and New Drugs: Are There New Perspectives in the Treatment of Anaemia?

**DOI:** 10.3390/jcm10040839

**Published:** 2021-02-18

**Authors:** Lucia Del Vecchio, Roberto Minutolo

**Affiliations:** 1Department of Nephrology and Dialysis, Sant’Anna Hospital, ASST Lariana, 22042 Como, Italy; 2Nephrology Division, Department of Advanced Medical and Surgical Sciences, University of Campania Luigi Vanvitelli, 80138 Naples, Italy; Roberto.MINUTOLO@unicampania.it

**Keywords:** anemia, chronic kidney disease, iron, erythropoiesis stimulating agents, PHD inhibitors, hypoxia inducible factor, hepcidin

## Abstract

Anemia is a well-known consequence of chronic kidney disease (CKD); it is mainly due to a relative insufficiency of erythropoietin synthesis by the failing kidneys. Over the years, the combination of erythropoiesis stimulating agents (ESA) and iron has become the standard of care of anemia. All ESAs effectively increase hemoglobin (Hb) levels in a substantial percentage of patients. However, in the last decade, their use has been surrounded by safety issues in increased cardiovascular risk, especially when used at high doses in inflamed and hyporesponsive patients. This has led to the definition of a more cautious Hb target. Iron deficiency is very frequent in CKD patients, with a higher frequency in non-dialysis patients. Traditionally, iron supplementation is mostly used as supportive therapy for anemia control. However, the concept is growing that intravenous iron therapy per se could be beneficial in the presence of heart failure. A new class of drugs, prolyl hydroxylase domain (PHD) inhibitors (PHD inhibitors) is becoming available for the treatment of anemia in CKD patients. Theoretically, these agents have a number of advantages, the main ones being that of stimulating the synthesis of endogenous erythropoietin and increasing iron availability. The impact of their future use in clinical practice is still to be defined. Another possible strategy could be targeting serum hepcidin and its related pathways. This possibility is fascinating from the scientific point of view, but at present its development phase is still far from clinical application.

## 1. Introduction

It was estimated that in 2015, chronic kidney disease (CKD) was affecting over 750 million persons worldwide [1]. This has important implications, since the disease has a strong negative impact not only on life expectancy, but also on disability-adjusted life-years (DALYs), as shown by a systematic analysis for the Global Burden of Disease Study in 2017 [2]. The increased risk for mortality and disability is not driven just by CKD itself, but by the comorbidities that either precede or follow CKD, namely cardiovascular (CV) disease, diabetes, and hypertension, or by its complications. 

Among others, anemia is a well-known consequence of CKD, with increasing severity and prevalence as CKD progresses. Its pathogenesis is mainly due to relative insufficiency of erythropoietin synthesis by the failing kidneys. However, it can be influenced by other causes, with inadequate iron availability and inflammation playing a prominent role.

Over the last decades, it has been clearly shown that the presence of anemia in CKD is associated with increased mortality and hospitalization risk. Moreover, it can significantly worse quality of life and increase the need for blood transfusions. Accordingly, the combination of erythropoiesis stimulating agents (ESA) and iron entered into clinical practice for the treatment of anemia in CKD patients: the first to stimulate inadequate erythropoiesis, the second to guarantee the availability of an essential constituent of the hemoglobin chains.

## 2. Current Evidence for the Use of Erythropoiesis Stimulating Agents 

Nowadays, ESA are an established treatment of anemia in CKD patients. Epoetin alfa and beta were the two first recombinant human erythropoietins entering into clinical practice nearly 30 years ago; some years later, darbepoetin alfa and methoxy polyethylene glycol-epoetin beta were developed. They differentiate from the two first ESAs for their longer half-life obtained by increasing glycosylation content and molecular size, respectively. More recently, biosimilars of epoetin alfa and beta have become available for clinical use. 

Despite the differences described above, all ESA molecules share a common mechanism of action, i.e., the activation of the erythropoietin receptor. This is expressed mainly on red blood progenitor cells, but also in several other tissues, where erythropoietin exerts pleiotropic effects other than erythropoiesis [3,4]. According to the ESA size and chemical structure, the bigger the molecule, the lower the receptor affinity, and the longer the half-life. This translates into different kinetics of receptor activation and internalization [5].

During these years of clinical use, ESA have been shown as effective agents in increasing and maintaining adequate hemoglobin (Hb) levels in a substantial proportion of CKD patients with a relatively acceptable safety profile. 

At the very beginning, the rationale for their use was quite simple: there was one symptom, anemia; there was a new effective drug to cure it, recombinant human erythropoietin; therefore, it was administered to increase Hb levels. Despite this apparent simplicity, a longstanding controversy took hold over the optimal Hb target to aim at when using ESAs. This went along with a more general change of perspective in medicine, since in many branches it became clear that improving one symptom or manifestation of a given disease did not necessarily imply a substantial advantage in terms of patient outcome.

Starting from the findings of several observational studies showing a better outcome in CKD patients with higher Hb levels [6,7,8], and on the wave of the enthusiasm over these new anti-anemic drugs, it was hypothesized that aiming at high, near-to-normal Hb levels with ESA treatment would have significantly improved patient outcome. Unfortunately, this hypothesis was not confirmed by randomized clinical trials. The first negative findings came from the “Normal Hematocrit Trial” at the end of the 1990s [9]. This study enrolled more than one thousand hemodialysis (HD) patients with clinical evidence of congestive heart failure or ischemic heart disease who were randomized to high (42%) vs. lower (30%) hematocrit levels. Although the difference in event-free survival between the two groups did not reach the prespecified statistical stopping boundary, the study was halted early because of a higher number of deaths and acute myocardial infarction in the group randomized to the higher hematocrit group. Data became more consistent after the publication of three other trials performed in non-dialysis CKD (ND-CKD) patients [10,11,12]. Again, randomization to a near-to-normal Hb value with ESA did not bring substantial benefits or could be even harmful, leading to a higher risk of arterial and vein thrombosis [13]. In particular, secondary analyses of the Trial to Reduce Cardiovascular Events With Aranesp Therapy (TREAT) [10] showed that in comparison to the placebo, randomization to darbepoetin alfa aimed at reaching Hb values of 13 g/dL led to a higher risk of stroke and possibly to increased risk of death for cancer in those with a previous history. Instead, anemia treatment with darbepoetin substantially reduced the need for blood transfusions in comparison to placebo. Post-hoc analyses of these trials showed that the patients not achieving the Hb target or requiring high ESA dose were exposed to a higher risk of cardiovascular events or death [14,15].

Altogether, this evidence led to the current clinical practice of ESA use, with a broad consensus on intentionally avoiding aiming at Hb levels > 13 g/dL with ESA therapy. A more cautious approach is now recommended, with a maximum Hb target of 11.5–12 g/dL, given by different guidelines/position papers [16,17]. A more individualized approach was also introduced, weighing the pros and cons of ESA therapy and the optimal Hb level to aim at based on comorbidities and risk factors. Particular caution is warranted in patients with specific risk factors (cancer, diabetes, symptomatic limb arteriopathy, stroke, or non-symptomatic ischemic heart disease). In these subjects, if ESA therapy is started, it seems wise to aim towards lower Hb levels [17]. Given concerns of an increased safety risk when using high-dose ESA therapy in hyporesponsive patients, there is wide consensus in avoiding excessive ESA dosing when ineffective and opting instead for blood transfusions when needed at low Hb levels.

Talking about the choice of ESA molecule in a given subject, different agents are usually prescribed based on administration convenience, CKD phase, administration route, and costs. Conversely, little attention is paid to the possibility that ESA molecules may have a different safety profile [18]. In a large, Japanese, registry-based cohort study of HD patients, long-acting ESA users had a 20% higher rate of all-cause death than short-acting ESA users in an unadjusted Cox-regression model [19]. The risk was higher for darbepoetin alfa users and among patients receiving high ESA doses. However, since this is an observational study, potential prescription biases at patient and facility level and residual confounders cannot be ruled out. Conversely, a pooled analysis of four observational studies of ND-CKD patients showed opposing findings, with an association between the use of a short-acting ESA with an increased risk of end-stage kidney disease (ESKD) or death in comparison with long-acting ESAs when given at higher doses [20]. However, a large, randomized, non-inferiority trial aimed at testing the risk of major adverse cardiovascular events or all-cause mortality in more than 2000 dialysis- and ND-CKD patients did not confirm possible safety issues of one single ESA molecule over the other [21]. Recently, Karaboyas et al. [22] analyzed the association between ESA molecule use and mortality in 65,706 hemodialysis patients of the DOPPS study across North America, Japan, and Europe. Overall, patients prescribed a long-acting ESA had a similar mortality rate to patients prescribed a short-acting ESA. Of note, a non-statistically significant trend for higher mortality risk was observed in Japanese patients receiving long-acting ESA, probably reflecting national prescription practices. 

## 3. The Benefits and Challenges of Iron Therapy

Iron is an essential component of oxygen-binding molecules (Hb and myoglobin) and it is required for the correct functioning of several cellular mechanisms, including DNA synthesis, enzymatic activities, and mitochondrial energy production [23,24]. Therefore, disturbances in iron metabolism, leading to iron deficiency (ID), have several clinical consequences. The most frequent is the development of iron-deficient anemia, but emerging evidence also supports its important involvement in cardiac diseases. Observational studies have evidenced that ID represents one of the most frequent complications in CKD patients, being detected in more than one half of the ND-CKD population [25,26,27,28] and 20–25% of dialysis patients [29]. This different prevalence may be explained by a different pattern of treatment; indeed, in CKD-ND patients, iron therapy is very frequently omitted and, when prescribed, it is mainly administered by the oral route [25,26], whose efficacy may be limited by its frequent gastrointestinal side effects and coexisting inflammatory state that strongly limits iron intestinal absorption [30]. As testified by large metanalyses, the response to oral supplementation is suboptimal in terms of ID correction and Hb increase in comparison with intravenous iron [31,32] even in trials excluding patients with intolerance to oral iron and with laboratory evidence of inflammation (C-reactive protein (CRP) >20 mg/L) [33]. Conversely, in the HD population, iron is exclusively administered intravenously, and at variance from CKD-ND, the majority of HD patients (70%) were treated with iron [29]. 

In the nephrology community, the use of iron has been almost exclusively adopted as supportive therapy for anemia control. Kidney Disease: Improving Global Outcomes (KDIGO) and European Renal Best Practice (ERBP) recommend iron use as a first-line strategy if an increase of Hb or a decrease in ESA dose is desired [16,17]. Similarly, almost all studies testing the effects of iron in CKD patients have been exclusively focused on the role of iron supplementation on anemia management by evaluating as primary outcomes either Hb increase, the achievement of Hb target, or ESA dose reduction [34]. Only two randomized clinical trials (RCTs) enrolling CKD patients have addressed the impact of intravenous iron on outcomes different from anemia correction. The Randomized trial to Evaluate intraVenous and Oral iron in chronic KidnEy disease (REVOKE) has compared intravenous iron sucrose versus oral iron on the rate of glomerular filtration rate (GFR) decline in ND-CKD [35], and the Proactive IV Iron Therapy in Hemodialysis Patients (PIVOTAL) trial has specifically assessed the effects of two different strategies of iron supplementation on all-cause mortality and incidence of non-fatal CV endpoints in HD patients [36]. The REVOKE trial randomly assigned 1:1 patients with CKD stages 3–4 and iron deficiency anemia to either open-label oral ferrous sulphate (325 mg three times daily for eight weeks) or intravenous iron sucrose (200 mg every two weeks, total 1 g); the primary endpoint was the decline in iothalamate-measured GFR over two years. This study has been prematurely stopped for futility because of little chance of finding differences in GFR slopes (between-group difference −0.35 mL/min/1.73 m^2^, (95%CI from −2.9 to +2.3)) and a higher risk of serious adverse events in the intravenous iron treatment group, in terms of cardiovascular events and infection episodes. These results have been largely debated because of incongruent safety signals in comparison with other RCTs and metanalyses [31,32,37,38,39,40].

More recently, the PIVOTAL trial compared in a large HD population (*n* = 2141) a proactive strategy of iron administration (iron sucrose 400 mg/month unless ferritin >700 ng/mL or transferrin saturation (TSAT) >40%) with a reactive strategy in which iron dosing (iron sucrose 100–400 mg/month) was driven by TSAT and ferritin levels [34]. Authors found that proactive administration of intravenous iron significantly reduced by 15% the risk of death and cardiovascular events (non-fatal myocardial infarction, non-fatal stroke, and hospitalization for heart failure). Mortality and cardiovascular risk protection were even higher when primary end-point components were analyzed as recurrent events (HR 0.77, 95% CI 0.66–0.92). 

The effects of iron therapy for correcting ID per se have been more extensively investigated in the cardiology setting and in particular in chronic heart failure (HF). In this high-risk population, the presence of ID is associated with increased mortality risk independently of the presence of anemia [41,42]. More importantly, RCTs demonstrated that intravenous iron supplementation in chronic HF with reduced ejection fraction (HFrEF) was effective in improving functional status and physical activity, as testified by the decrease of NT-proBNP levels [43], the increase of peak oxygen consumption [44,45], the reduction of NYHA class [46], and the increase of 6-min walking distance [47]. Most of these trials also reported an improvement of quality of life among secondary outcomes. Additionally, a post-hoc analysis of the RCT Myocardial-IRON in 53 HF patients with ID receiving ferric carboxymaltose (FCM) versus placebo reported a significant short-term increase in left ventricular function (30 days after FCM: ejection fraction +4.1%, *p* = 0.01) and right ventricular ejection fraction (+3.2% and +4.7% at day 7 and day 30 post-FCM, *p* < 0.005) assessed by cardiac magnetic resonance [48]. Finally, a patient-level metanalysis of four placebo-controlled RCTs including 839 HFrEF patients receiving FCM showed a significant reduction in the risk of hospitalizations for CV and CV mortality (HR 0.59, 95% CI 0.40–0.88) [49]. Intravenous iron has recently been tested also in patients with acute HF and ID (*n* = 1110) by showing that after hospital discard for an episode of acute HF, treatment with FCM reduced the risk of re-hospitalizations by 26% without significant effect on the risk of cardiovascular death [50].

Of note, available studies suggesting an improvement in symptoms and patient-reported outcomes with the use of intravenous iron in HF patients with ID should be corroborated by the evidence that this treatment also has an impact on hard endpoints before it can be widely incorporated into clinical practice. Nonetheless, while waiting for larger studies assessing the effects of intravenous iron on mortality, morbidity, and hospitalization, current guidelines for HFrEF do recommend with a strong level of evidence the correction of ID with intravenous iron independently from the Hb level to alleviate symptoms and improve exercise capacity and quality of life [51,52]. These cardiology guidelines, at variance with nephrology ones, do not include the presence of low Hb levels as a modifying factor of the therapeutic approach and do not focus on the effectiveness of iron therapy on anemia correction.

Experimental studies using knockout transgenic models for different components of iron metabolism have shed some light on the pathophysiological link between disturbed iron metabolism in cardiomyocytes and functional/structural abnormalities of the heart [53]. It has been also demonstrated that iron-depleted cultured cardiomyocytes displayed impaired mitochondrial ATP-linked respiration inducing a significant reduction of contractile force and maximum contractile and relaxation velocity that fully recovered after the addition of transferrin-bound iron to cell cultures [54]. The presence of an impaired mitochondrial function as a consequence of reduced myocardial iron content has also been demonstrated in a study comparing left ventricular samples obtained from consecutive HF patients undergoing transplantation (*n* = 91) with organ donors (controls, *n* = 38) [55]. Authors found that lower iron content in cardiomyocytes from HF patients associates with reduced mitochondrial activity and reduced expression of reactive-oxygen species (ROS)-protective enzymes, thus suggesting that in iron-deficient HF patients, impaired energy production, as well as reduced ROS protection, may promote contractile dysfunction and maladaptive remodeling. 

Mitochondrial dysfunction is also described in CKD patients. Gomboa et al. reported a significant alteration of mitochondrial structure (volume density) and function (mitochondrial DNA copy number) in skeletal muscle biopsies obtained from 10 HD patients in comparison with 17 controls [56]. The same authors also evaluated in vivo mitochondrial function at the level of knee extensors using ^31P^magnetic resonance spectroscopy to obtain the phosphocreatine recovery time constant (a measure of mitochondrial function) in three groups of subjects (controls *n* = 21, CKD-ND 3–5 *n* = 20, and HD *n* = 22) [57]. They reported a prolonged phosphocreatine recovery constant in HD patients that was already detected in those with CKD-ND. Furthermore, they found a strong and significant relationship between mitochondrial dysfunction and physical performance assessed by a 6-min walking test (r = 0.62, *p* < 0.001). 

These findings could theoretically represent a good rationale for considering the usefulness of ID correction in CKD patients not only aimed at anemia correction, but also at improving cardiac performance, physical activity, and quality of life. Whether the adoption of the “cardiologic” approach based on an “iron first” strategy also in the CKD population would have an impact on improving outcomes remains to be demonstrated. However, moving from a concept of “supplementation” to that of “treatment” would certainly allow reducing the large therapeutic inertia for ID and optimizing and postponing ESA therapy. 

## 4. PHD Inhibitors for the Treatment of Anemia

Prolyl hydroxylase domain (PHD) inhibitors are a new class of drugs for the treatment of anemia. They differentiate from ESAs, since they do not directly activate the erythropoietin receptor, but rather stimulate the production of endogenous erythropoietin from the kidneys and to a lesser extent from the liver. Moreover, they are administered orally and not parenterally. 

The process of development that brought to the synthesis of these molecules started nearly 25 years ago from the basic science studies of three researchers, who recently received the Nobel prize in recognition of their work. In their seminal papers [58,59], they firstly described hypoxia-inducible factor 1 (HIF-1) as an essential transcriptional factor to activate the erythropoietin gene enhancer in hypoxic cells (Figure 1). 

The HIF system is a complex pathway that orchestrates many other physiological functions other than erythropoiesis based on tissue O_2_ content. Indeed, in the case of hypoxia, the system promotes several actions to increase O_2_ supply to cells and preserve their viability. This involves the activation of multiple genes regulating erythropoiesis, iron metabolism, angiogenesis, lipid and glucose metabolism, glycolysis, mitochondrial function, inflammation and immunity, cell growth and survival, vasodilation, and cell migration [60].

PHD enzymes tightly regulate the availability of the HIF-α subunit based on O_2_ and iron availability; in the case of hypoxia, they are less active; do not start HIF-α subunit degradation, which in turn translocates to the nucleus; heterodimerize with the β subunit; and act as a functional transcription factor. Therefore, PHD inhibitors simulate the effect of hypoxia on the HIF system. This simple statement in reality is much more complex than that, since the degree and duration of hypoxia bring different and often opposing biological effects. Similarly, the potency and spectrum of inhibition of single molecules of PHD inhibitors could have not necessarily overlapping effects. Overall, it seems that PHD inhibitors act as mild hypoxia.

Several PHD inhibitors have been undergoing a large clinical development phase for the treatment of anemia in CKD patients; at present, four molecules of the class, roxadustat, vadadustat, daprodustat, and enarodustat, have received marketing authorization either in China or in Japan or both.

According to data from phase-II studies, compared with placebo or ESA, PHD inhibitors are effective in increasing and maintaining Hb levels in both ND and dialysis-dependent CKD patients with a satisfying safety profile in the short-term [61,62,63,64,65,66,67].

Similar findings were obtained during phase-III RCTs, as confirmed by the results of a recent metanalysis including both phase-II and phase-III RCTs [68,69].

Table 1 summarizes phase-III clinical studies that have been published so-far in full.

Chen et al. reported the data of two RCTs performed in China in 154 ND-CKD [63] and in 305 dialysis-dependent patients [64], who were randomized to either roxadustat or placebo (for the ND-CKD population) or the comparator ESA in dialysis in a 2:1 ratio. The starting dose of roxadustat was given thrice weekly: 100 mg for dialysis patients weighing 45–60 kg, or 120 mg for those weighing >60 kg, and 70 or 100 mg according to the same weight cut-off for ND-CKD patients. In ND-CKD patients, roxadustat significantly increases Hb levels from baseline (mean change of 1.9 ± 1.2 g/dL) during the 26-week treatment period; as expected, patients receiving placebo experienced an opposing trend (−0.4 ± 0.8 g/dL) [63]. In dialysis patients, roxadustat was found non-inferior to epoetin alfa on the change in Hb level from baseline, but with a greater mean change (0.7 ± 1.1 g/dL and 0.5 ± 1.0 g/dL, respectively) [64]. The difference was more remarkable in the first 12 weeks of treatment, possibly suggesting a greater potency of the roxadustat starting dose in comparison to that of epoetin alfa.

In both studies, hyperkalemia and metabolic acidosis occurred more frequently in the roxadustat group.

Recently, Akizawa et al. reported the findings of two, non-comparative, randomized phase- III studies, which were performed in Japan [70]. Differing from the studies by Chen et al. [63,64], roxadustat starting dose was either 50 or 70 mg thrice a week according to randomization in the 75 ESA-naïve patients or 70 or 100 mg based on prior ESA dose in 164 ESA-converted patients. The proportion of patients who achieved an average Hb of 10.0–12.0 g/dL was 73.0% in the ESA-naïve patients and 79.1% in the ESA-converted ones. The suboptimal efficacy is possibly due to the lower starting dose of these two trials. In hemodialysis Japanese patients, roxadustat also met non-inferiority in comparison to darbepoetin alfa in the mean change of Hb from baseline [71].

**Table 1 jcm-10-00839-t001:** Phase-III clinical studies of PHD inhibitors published as full papers.

Study	Design	Country	Patients	Drug	Dose	Comparator	Main Effect	Other Effects	Follow-Up
Chen et al. [63]	Double-blind then open label	China	154, ND	Roxadustat	70 mg < 60 kg body weight or 100 mg for ≥60 kg body weight 3 times a week	Placebo	Mean Hb change from baseline of 1.9 ± 1.2 g/dL with roxadustat and 0.4 ± 0.8 g/dL with placebo	Decrease of serum hepcidin and serum cholesterol	8 + 18 weeks
Chen et al. [64]	Open-label, active-controlled	China	305, dialysis	Roxadustat	100 mg < 60 kg body weight or 120 mg for ≥60 body weight 3 times a week	Epoetin alfa	Non-inferiority met: mean ΔHb change from baseline of 0.7 ± 1.1 g/dL with roxadustat and 0.5 ± 1.0 g/dL with epoetin alfa	Decrease of serum hepcidin, lower decrease of TSAT, decrease of serum cholesterol	26 weeks
Akizawa et al. [71]	Double-blind, double-dummy	Japan	303, HD	Roxadustat	70 mg or 100 mg	Darbepoetin alfa	Non-inferiority met: mean of Hb from baseline was −0.04 g/dL (95% CI, −0.16 to 0.08 g/dL) and −0.03 g/dL (95% CI, −0.14 to 0.09 g/dL) for roxadustat and DA, respectively, with the estimated difference of −0.02 g/dL (95% CI, −0.18 to 0.15 g/dL)	No remarkable changes in the mean hepcidin values in the two treatment groups; the use of oral and IV iron was gen- erally similar during the study. A trend was seen for higher median doses of DA in those with hs-CRP ≥ 3000 mg/L	24 weeks
Akizawa et al. [70]	Non-comparative, randomized	Japan	75 ESA-naïve, dialysis	Roxadustat	50 or 70 mg thrice a week	None	73.0% of patients achieving average Hb 10.0–12.0 g/dL	Hepcidin decrease	24 weeks
Akizawa et al. [70]	Non-comparative, randomized	Japan	164 ESA-converted, dialysis	Roxadustat	70 or 100 mg thrice a week	None	79.1% of patients achieving average Hb 10.0–12.0 g/dL	Hepcidin decrease	52 weeks
Akizawa et al. [72]	Non-comparative, randomized	Japan	100, ND-CKD	Roxadustat	50 or 70 mg three times weekly	None	97.0% (CI 91.4, 99.4) achieving Hb ≥ 10.0 g/dL and 94.9% (CI 88.6, 98.3) achieving Hb ≥ 10.5 g/ dL	Hepcidin decrease	24 weeks
Akizawa et al. [73]	Randomized, double-blind, active-control	Japan	271, HD	Daprodustat	4 mg/day	Darbepoetin alfa	Non inferiority; mean Hb during weeks 40–52 within the target range in both groups (10.9 g/dL [95% CI, 10.8 to 11.0] for daprodustat, and 10.8 g/dL [95% CI, 10.7 to 11.0] for darbepoetin alfa.	Higher hepcidin decrease with daprodustat Broader range of darbepoetin alfa dose in comparison to daprodustat according to ERI categories	52 weeks
Nangaku et al. [74]	Single-arm	Japan	42, PD	Vadadustat	300 mg/day	None	Mean of average Hb at weeks 20 and 24 of 11.35 g/dL (within the target range)	NA	24 weeks

ND: non-dialysis; Hb: hemoglobin; TSAT: transferrin saturation; CI: confidence interval; ESA: erythropoiesis stimulating agents: CKD: chronic kidney disease; HD: hemodialysis; PD: peritoneal dialysis; NA not available; DA: darbepoetin alfa; hs-CRP: high-sensitive C reactive protein.

Another Japanese, phase-III, non-comparative RCT was conducted in 100 ND-CKD patients, who were randomized to an initial dose of roxadustat of either 50 or 70 mg administered three times weekly [72]. As expected, in this non-dialysis population, the response rate was higher than that observed in ESA-naïve dialysis patients treated with the same dose [70]. Interestingly, at week 24, the maintenance dose of roxadustat was of 36 mg thrice a week in both dose groups. 

Data of a phase-III RCT with daprodustat were also published [73]. In this double-blind study, 271 Japanese, HD patients were randomized to either daprodustat 4 mg/day or darbepoetin alfa based on the previous ESA dose. Daprodustat was found non-inferior to darbepoetin alfa in maintaining Hb levels during 40–52 weeks. Similarly, as it has been observed with roxadustat, mean hepcidin levels decreased more with daprodustat than with darbepoetin alfa; this went together with an increase in total iron-binding capacity only in the daprodustat group.

Phase-III data of vadadustat have been published in full only for peritoneal dialysis patients [74]. In this single-arm study, 42 Japanese patients received vadadustat at a starting dose of 300 mg/day for 24 weeks. During the follow-up, Hb levels remained in the pre-specified target range, without significant safety issues.

At the time of writing this article, several other large RCTs with PHD inhibitors have been concluded; their findings have been published only in abstract forms or communicated with press releases. Overall, roxadustat and vadadustat showed efficacy in terms of Hb increase or maintenance in comparison to placebo or the comparator ESA with a good safety profile. Focusing on major adjudicated cardiovascular events (MACE), the cardiovascular safety profile was in general non-inferior to the control groups with two exceptions of opposing nature (a reduced hazard ratio for roxadustat in incident dialysis patients in comparison to placebo, but a higher one for vadadustat in ND-CKD patients in comparison to darbepoetin alfa) [30,75]. The reason for these discrepancies is unclear. Data for the ASCEND program with daprodustat are not available yet.

Similar to ESAs, HIF-PHD inhibitors also reduce the need for blood transfusions [76]. 

Apart from the stimulation of erythropoiesis, PHD inhibitors have additional effects that could be helpful in CKD patients and possibly overcome some of the drawbacks of ESA.

As anticipated above, the HIF system is involved in the regulation of iron metabolism to increase its availability in conditions of hypoxia and thus increase erythropoiesis. Accordingly, several data suggest a higher degree of iron availability during PHD inhibitor therapy at similar erythropoietic stimuli, as testified by a greater extent of hepcidin decreases. However, available data have several confounding factors, in particular different rates of Hb correction and non-standardized iron therapies.

Coherently with the knowledge that the HIF pathway plays an important role in adaptation to inflammation [77], it seems that the erythropoietic response of PHD therapy is less influenced by inflammation in comparisons to ESA therapy [64]. 

Recently, Akizawa et al. [78] performed a post-hoc analysis of a phase-3 RCT of Japanese HD patients to assess the impact of factors associated with ESA hyporesponsiveness on roxadustat and darbepoetin alfa dose needs. A significant increase in mean weight-adjusted dose needs at six weeks was observed for both roxadustat and darbepoetin alfa with increasing erythropoiesis responsive index (ERI), with a non-statistically significant trend of higher dose change with increasing ERI values for darbepoetin than roxadustat. Conversely, at lower iron repletion markers, dose needs remained stable for Roxadustat, but increased for darbepoetin alfa. Finally, roxadustat doses were less affected by hs-CRP levels. 

Another interesting ancillary effect of some PHD is that they decrease serum total cholesterol; the same is not observed in the epoetin alfa and placebo groups [63,64]. Of note, the effect is not limited only to total serum cholesterol and LDL particles, but also to HDL. It is unknown whether these changes translate into CV benefits.

No clinically relevant safety issues have been shown so far with PHD inhibitors. However, the HIF system controls a wide array of pathways, some of them still unknown. This may open the way to new ancillary positive effects, but also to unwanted adverse events. The most feared ones are the increase of vascular endothelial growth factor, leading to increased cancer risk and worsening of diabetic retinopathy. Of note, ophthalmological examinations of dialysis patients did not show an increased risk of retinal hemorrhages or increased retinal thickness abnormalities in those randomized to roxadustat in comparison to darbepoetin alpha over 24-weeks [79]. Other possible complications include pulmonary hypertension and cyst growth in patients with polycystic kidney disease. Moreover, some clinical studies have shown a higher frequency of mild hyperkalemia and metabolic acidosis, the first as a likely consequence of the second. Interestingly, the HIF system has several opposing actions on the acid–base homeostasis [80]. From one side, HIF activation increases glucose uptake and glycolysis, causing lactate accumulation via anaerobic glycolysis, and promotes extracellular acidification via increased conversion of CO_2_ to bicarbonate and proton by the intracellular carbonic anhydrase IX, followed by proton excretion outside the cell. Conversely, HIF enhances liver gluconeogenesis from circulating lactate and protects against lactic acidosis. 

## 5. New Drugs Targeting the Hepcidin Pathway

Inflammation and elevated hepcidin with suppressed erythropoiesis and reduced iron availability are accompanying features of many patients with anemia and CKD. For this reason, drugs directly targeting hepcidin and disrupting its function have been proposed as possible future treatment strategies [81]. A spiegelmeier of the molecule was synthetized some years ago; however, its clinical development did not pass phase II. Humanized monoclonal antibodies targeting hepcidin have been also developed by Amgen (12B9m) and Eli Lilly (LY2787106); their clinical development did not start or was terminated early. Anticalin proteins that antagonize hepcidin function seem more promising. These are a class of small proteins with designed ligand-binding properties and derived from the natural human lipocalins. PRS-080 (Pieris Pharmaceuticals, Inc, Boston, MA, USA) is a pegylated anticalin protein that antagonizes hepcidin. Phase-I data showed that the drug was safe and well-tolerated in healthy volunteers and HD patients [82]. Data of further development are not available at the moment.

The regulation of hepcidin synthesis is complex, with both positive and negative regulators. Some of them have become potential targets of new drug development to treat anemia during inflammation. The most important is that involving bone morphogenetic proteins (BMPs) and the intracellular transducing SMAD molecules. BMP are potent stimulators of hepcidin production. Modified heparins with lower anticoagulant properties, the so-called “glycol-split” heparins, can directly bind to BNP, hamper its function, and consequently decrease hepcidin production. They also have antitumor effects in multiple myeloma. Some of these molecules showed anemia improvement in experimental models [83]. Pentosan polysulfate is a drug with low anticoagulant activity and a high degree of sulfation similar to that of heparin; it is already in clinical use for the treatment of interstitial cystitis and osteoarthritis. Interestingly, the drug was shown to decrease serum hepcidin in a mouse model [84]. Soluble hemojuvelin-Fc fusion protein (sHJV.Fc) can inhibit BMP signaling; an exploratory study with FMX-8 was conducted some years ago to treat anemia in CKD patients but was terminated early for recruitment difficulties (NCT02228655). Several other compounds act as BMP receptor inhibitors and BMP coreceptor inhibitors. Among them, TP-0184 (Tolero Pharmaceuticals, Inc, Salt Lake City, UT, USA) is now under evaluation in a phase I–II study of patients with myelodysplastic syndrome of low or intermediate risk (NCT04623996).

Another regulator of hepcidin synthesis is interleukin 6 (IL-6). Interestingly, tocilizumab, a humanized anti-IL-6 receptor antibody, improves anemia and reduce serum hepcidin in patients with rheumatoid arthritis [85]. Recently, ziltivekimab, a novel anti-IL-6 ligand antibody, was tested in phase I/II randomized clinical trial of 61 HD patients with a single nucleotide polymorphism of the TMPRSS6 gene, which is known to increase the effects of IL-6 on inflammation, elevated IL-6, and hyporesponse to ESA therapy [86]. In comparison to placebo, ziltivekimab obtained a significant decrease in inflammation and iron parameters, including serum hepcidin. A decrease of median ESA dose with increasing ziltivekimab dose was also observed in parallel with an increase of Hb levels. During follow-up, four patients died in the ziltivekimab group; two of them had sepsis. This raises possible questions of safety when using drugs interfering with inflammation mechanisms. A phase-II study with ziltivekimab in patients with advanced CKD (RESCUE-2) has just started requitement in Japan (NCT04626505). 

IL-6 effect is mediated by the downstream JAK-STAT pathway. Several JAK2-STAT3 inhibitors have been shown to reduce hepcidin and improve anemia in vitro or in animal models; others have been developed as anticancer agents and possibly have anti-hepcidin effects. However, they are not likely to enter clinical use for the treatment of anemia.

Finally, anti-ferroportin antibodies have been developed; these agents prevent the binding of hepcidin to ferroportin without altering its function. One of these antibodies, LY2928057 (Eli Lilly), showed efficacy in CKD patients in terms of better Hb levels and lower serum ferritin in comparison to placebo [87]. However, further studies have not been planned.

## 6. Future Perspective

The treatment with either ESA or iron or their combination is the standard of care of anemia in CKD patients. Even if they have been proven effective over the years, both treatment strategies have some drawbacks, especially in subsets of patients who do not achieve optimal correction of anemia despite receiving high doses of one or both treatments.

The availability of new anti-anemic agents with a different mechanism of actions to that of ESA is bringing new hope for obtaining adequate anemia correction/maintenance in a higher percentage of patients and possibly with an improved safety profile.

Theoretically, PHD inhibitors could bring advantages in comparison to ESA. First of all, they stimulate the production of endogenous erythropoietin and for similar efficacy/erythropoietic stimulus expose patients to lower concentrations peak. Second, they seem to be effective also in inflamed patients and reduce dose needs in this high-risk population. Third, they improve iron utilization; this could be of importance, especially in inflamed patients who usually have functional ID.

Preliminary data of phase-III studies showed non-inferiority rather than superiority in reducing MACE risk in comparison to ESAs and no significant MACE risk reduction in comparison to placebo; this has somehow cooled expectations over these new class of drugs. Regardless their complex and widespread mechanism of action still leave us with the idea that we have to learn much more from clinical and experimental studies.

## Figures and Tables

**Figure 1 jcm-10-00839-f001:**
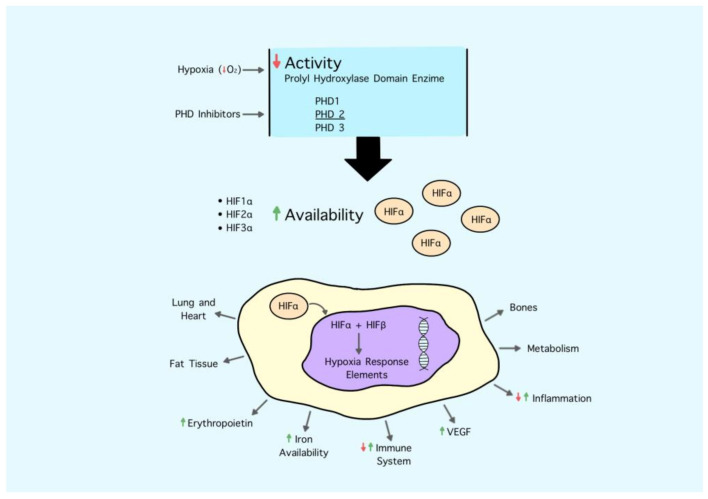
Hypoxia inducible factor (HIF) is a heterodimer composed of oxygen-sensitive HIFα and constitutively expressed HIF-β subunits. Under normal conditions, the HIF-α subunit is rapidly degraded by prolyl hydroxylation domain protein (PHD). During hypoxia or following treatment with PHD inhibitors, PHD is less active and thus more HIF- α is available to translocate into the nucleus. Here it recruits HIF-1β and induces the expression of specific target genes, including those leading to the synthesis of erythropoietin and controlling iron metabolism.

## Data Availability

Not applicable.

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
