# Peer review of "ESA, Iron Therapy and New Drugs: Are There New Perspectives in the Treatment of Anaemia?"

_jcm, 2021, doi:10.3390/jcm10040839_

Round 1

Reviewer 1 Report

This is a timely update of current evidence for anaemia management in patients with CKD/ESKD, with a focus on PHD inhibitors.  Current evidence on ESA, iron supplementation and PHD inhibitors is comprehensively reviewed.

I have no major suggestions to improve the manuscript.

Minor suggestions:

  1. Given the focus on PHD inhibitors, a figure to illustrate their mechamism of action would be helpful
  2. Whilst the manuscript is clearly written with a logical flow, it would benefit from some English language editing throughout.

Author Response

Thank you for your suggestions.

We have considered them in the new version of the manuscript

Reviewer 2 Report

This is a thorough and well-presented review on the use of erythropoiesis stimulating agents and the latest generation of PHD inhibitors. This review is well written, and can be read by average readers without much effort. I recommend only very few suggestions.

Specific comments:

L 260: ‘which in turn becomes available to enter the cell’: change into: ‘which in turn translocates to the nucleus’ (they are already in the cell)

L 261: and act as a functional transcription factor

L 290: hyperkalemia and metabolic acidosis as side effects of roxadustat:  It would be highly desirable if the authors expand this paragraph and could comment more on the potential PHD inhibitor-dependent mechanism (even if speculative) of these worrying possible side effects.

Typo’s:

L 224: Same authors

L 355: the HIF system controlS

L 357: the most fEAred ones

Table1 foot note: CKD: chronic

Author Response

Thank you for your suggestions

We took care of all them

Reviewer 3 Report

This is an excellent review on the current treatment of anemia and what to expect with the new therapies. The manuscript is well written specially the historical part on ESA

I will have some suggestions to improve the review:

1/ Regarding the controversy on the safety profile of ESA after the report of sakaguchi and coll, the authors should also cite the recent paper of Karaboyas and coll (KI report 2020). Briefly, using the DOPPS dataset, they confirmed the result of sakaguchi in the Japanese population but not in the European, nor north American population. This finding probably reflects the difference in national practice. What is true in Japan is true in Japan. Japan has the lowest mortality for HD patient as compare to the rest of the world.

2/ From a cardiologist point of view; the authors should emphasize that ID is also frequent (50% of heart failure Klip American journal of heart failure 2013) . Mechanism of ID in Heart Failure is roughly similar as in CKD and a majority of patient with HF have CKD. Thus, nephrologist should probably consider ID treatment to correct anemia but also to improve the cardiac function. 

3/ In PID part:  the authors should include the latest study by Provenzano and coll. It is a pooled analysis of 3 studies on incident HD patient treated with Roxadustat. Their results suggest that PID are associated with less Cardiovascular events as compared to epoetin-alfa. It should be note that some concerns such as worsening of diabetic retinopathy has been challenged (Sepah Yasir Oral communication ERA-EDAT 2020)

4/ I will suggest a new paragraph in the perspective regarding to the management of inflammation in anemia. There are multiple drugs on phase I/II/III that target the hepcidin pathway. There is clearly a new field of investigation and treatment that will emerge soon. The latest “Ziltivekimab for Treatment of Anemia of Inflammation in Patients on Hemodialysis” published recently in JASN should be cited.

Author Response

Thank you for your suggestions.

In particular:

-we quoted and commented the paper  Karaboyas and coll 

-we added a comment that nephrologists should treat ID also from the cardiological perspective

-we included the latest study by Provenzano and coll and added a comment on diabetic retinopathy

-we added a small new chapter on  drugs targeting the hepcidin pathway and quoted the paper “Ziltivekimab for Treatment of Anemia of Inflammation in Patients on Hemodialysis”